# Attrition and associated factors among children living with HIV at a tertiary hospital in Eritrea: a retrospective cohort analysis

Samuel Tekle Mengistu ,[1,2] Ghirmay Ghebrekidan Ghebremeskel ,[1,2] Aron Rezene,[3] Mahmud Mohammed Idris,[4] Tsegereda Gebrehiwot Tikue,[4] Mohammed Elfatih Hamida ,[5] Oliver Okoth Achila [6]

[1]General Medicine, Nakfa Hospital, Ministry of Health Northern Red Sea Branch, Nakfa, Eritrea
[2]Medicine, Orota School of Medicine and Dentistry, Asmara, Eritrea
[3]Maternity Health, Edaga Hamus Hospital, Ministry of Health Maekel Branch, Asmara, Eritrea
[4]Department of Pediatrics and Child Health, Orotta College of Medicine and Health Sciences, Asmara, Eritrea
[5]Unit of Medical Microbiology, Orotta College of Medicine and Health Sciences (OCMHS), Asmara, Eritrea
[6]Unit of Clinical Laboratory Science, Orotta College of Medicine and Health Sciences (OCMHS), Asmara, Eritrea

**Correspondence to**
Dr Samuel Tekle Mengistu;
teklesam7@gmail.com

## ABSTRACT

**Background** Reducing attrition in paediatric HIV-positive patients using combined antiretroviral therapy (cART) programmes in sub-Saharan Africa is a challenge. This study explored the rates and predictors of attrition in children started on cART in Asmara, Eritrea.

**Methods** This was a retrospective cohort study using data from all paediatric patients on cART between 2005 and 2020, conducted at the Orotta National Referral and Teaching Hospital. Kaplan-Meier estimates of the likelihood of attrition and multivariate Cox proportional hazards models were used to assess the factors associated with attrition. All p values were two sided and p<0.05 was considered statistically significant.

**Results** The study enrolled 710 participants with 374 boys (52.7%) and 336 girls (47.3%). After 5364 person-years' (PY) follow-up, attrition occurred in 172 (24.2%) patients: 65 (9.2%) died and 107 (15.1%) were lost to follow-up (LTFU). The crude incidence rate of attrition was 3.2 events/100 PY, mortality rate was 2.7/100 PY and LTFU was 1.2/100 PY. The independent predictors of attrition included male sex (adjusted HR (AHR)=1.6, 95% CI: 1 to 2.4), residence outside Zoba Maekel (AHR=1.5, 95% CI: 1 to 2.3), later enrolment years (2010–2015: AHR=3.2, 95% CI: 1.9 to 5.3; >2015: AHR=6.1, 95% CI: 3 to 12.2), WHO body mass index-for-age z-score <−2 (AHR=1.4, 95% CI: 0.9 to 2.1), advanced HIV disease (WHO III or IV) at enrolment (AHR=2.2, 95% CI: 1.2 to 3.9), and initiation of zidovudine+lamivudine or other cART backbones (unadjusted HR (UHR)=2, 95% CI: 1.2 to 3.2). In contrast, a reduced likelihood of attrition was observed in children with a record of cART changes (UHR=0.2, 95% CI: 0.15 to 0.4).

**Conclusion** A low incidence of attrition was observed in this study. However, the high mortality rate in the first 24 months of treatment and late presentation are concerning. Therefore, data-driven interventions for improving programme quality and outcomes should be prioritised.

## WHAT IS ALREADY KNOWN ON THIS TOPIC

⇒ Studies in low/medium-income countries of sub-Saharan Africa and Asia have demonstrated that retention of children living with HIV in care is a major challenge. Multiple evaluations have suggested that in the context of the rapid scale-up of combined antiretroviral therapy, reducing attrition (lost to follow-up (LTFU) and mortality) rate is a key programmatic priority.

## WHAT THIS STUDY ADDS

⇒ A quarter of the patients were not retained in care.
⇒ Attrition in the first 2 years after enrolment was largely due to death.
⇒ Attrition was mostly associated with enrolment in recent years, malnutrition, advanced HIV disease (III or IV) at enrolment, initiation of backbones other than zidovudine+lamivudine and residence outside Zoba Maekel.

## HOW THIS STUDY MIGHT AFFECT RESEARCH, PRACTICE OR POLICY

⇒ Our findings emphasise on the need of linked databases across the HIV/AIDS care cascade within Eritrea.
⇒ It also highlights the importance of enhanced outreach and intensive case finding/tracing for LTFU, as standard components of HIV/AIDS care programmes in Eritrea are equally evident.

## INTRODUCTION

Despite advancements in HIV management, HIV infection in children remains a major public health burden. A recent Joint United Nations Programme on HIV/AIDS (UNAIDS) report suggested that of the 37.7 million (30.2–45.1 million) people living with HIV (PLWH) in 2020, children <14 years account for ~1.8 million (1.3–2.2 million).[1] In the same period, 180 000 children were newly infected and 110 000 children died of HIV/AIDS. Importantly, a disproportionate number of these children were from sub-Saharan Africa (SSA), where resources are limited and the burden of infectious disease is high.

The prevailing situation undermines the aspirational UNAIDS 90-90-90 target to end the HIV/AIDS epidemic by 2030 (Project 2030).[1] Although combined antiretroviral therapy (cART) helps PLWH live long and healthy lives, and marked improvements have been registered in almost all principal landmarks in the HIV care cascade globally, empirical data from low/medium-income countries (LMICs) in SSA and Asia suggest that children living with HIV (CLHIV), particularly young adolescents, still experience increasing HIV/AIDS-related mortality.[2] Multiple studies in LMIC settings also demonstrated that a large proportion of CLHIV continue to present to care centres with advanced HIV disease, and that retention is a major challenge.[2–4] Hence, emerging reports from the region point (sensu stricto) to significant problems in the fast-track UNAIDS 90-90-90: diagnosis (early)–treat (promptly)–retain (to ensure viral suppression) paradigm, thereby putting targeted timelines off track.[1]

Responses to this problem call for precise rapid scale-up of cART and reduction of attrition (lost to follow-up (LTFU) and mortality) in LMIC HIV/AIDS programmes. This suggestion is based on data showing that attrition is high among younger children (<14 years), approximately 14% at 12 months' follow-up and 22% at 36 months.[5 6] Furthermore, some experts argue that reducing attrition rates is not only necessary for continued cART, but it also allows clinicians to evaluate the emergence of medication toxicity. It also gives them the opportunity to prevent, manage or treat opportunistic infections, detect treatment failure and initiate prompt switching of cART. Additional benefits include ancillary services such as social support and counselling. In this regard, studies on attrition and associated factors among CLHIV in jurisdictions across SSA should be a principal research concern in the foreseeable future.

Therefore, we aimed to describe the incidence of attrition and associated factors among CLHIV in Asmara, Eritrea, a country with one of the lowest HIV prevalence in SSA.[7] Data gathered from this study can be used as a benchmark for gauging successes and weaknesses of programmes,[5] and to identify critical gaps and opportunities for context-specific interventions aimed at minimising attrition and maximising retention. We also explored the multiple temporal trends associated with programme performance, mortality and enrolment rates.

## METHODOLOGY
### Study design and setting
This retrospective cohort study was conducted in the paediatric HIV/AIDS follow-up clinic at the National Pediatric Referral Hospital (NPRH). The HIV/AIDS follow-up clinic in NPRH was commissioned in 2005, making it the first institution in Eritrea to offer cART for CLHIV. Before the decentralisation of services to other zones (2010), NPRH (in the Maekel zone) was the only institution in the country offering cART to CLHIV. In total, 822 children aged below 15 years received service

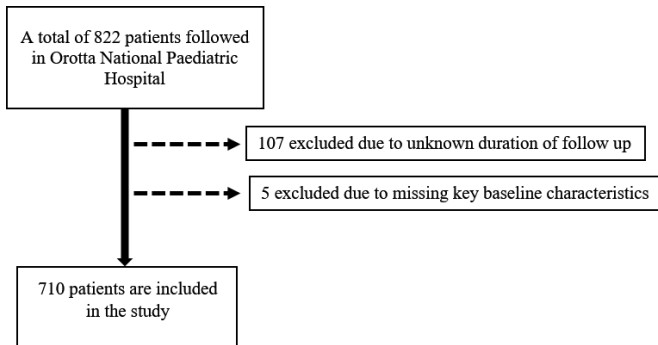

**Figure 1** Flow chart for study participant recruitment.

and/or had been enrolled at the clinic since their inception.

### Data collection and approach
We used the data collected from the database of the Orotta National Referral and Teaching Hospital paediatric cART and follow-up programme (cART District Health Information System 2 [DHIS 2]). The retrieved data were robustly reviewed and cross-referenced with patient clinical chart registries. The data retrieved from this repository were exported to Microsoft Excel. Baseline data collected included the date of birth/age at enrolment, sex, address, cART initiation date, initial cART regimen, presence of tuberculosis (TB), baseline WHO clinical stage, baseline CD4 count, weight, haemoglobin concentration and complete blood count. Height is not available in electronic medical records and was thus retrieved from clinical chart registries. The WHO SDs for weight for age, height for age and body mass index (BMI) were computed using the WHO SD for emergency nutritional assessment software (www.nutrisurvey.net/ena/ena.html). The recruitment details are illustrated in figure 1.

### Operational definitions
- Death was defined as deaths occurring after enrolment in follow-up but before 31 December 2020.[8–10]
- Patients were considered as LTFU if they had missed scheduled visits for ≥3 consecutive months.[8]
- Attrition was defined as LTFU or death during follow-up in the clinic.[5 11]
- Retention was defined as children who were alive and on follow-up in the clinic at the end of the study period and children who were transferred out at any time during the study period.[11]
- Advanced HIV disease was defined as a CD4 cell count <200 cells/mm³ and/or WHO stage 3 or 4 events for patients >5 years of age.[12]
- Late presentation was defined as patients enrolled with advanced WHO HIV stage (III/IV) and/or T-cell CD4⁺ count <200 cells/μL.
- Multiple measures were used to measure the nutritional status. These included the weight-for-age z-score (WAZ), height-for-age z-score (HAZ) and

BMI-for-age z-score (BAZ). For nutrition, BAZ underscores <−2 SD, WAZ ≤−2 and/or HAZ ≤−2.

► Residence outside Zoba Maekel was defined as children who reside outside Zoba Maekel, the administrative zone of Eritrea.

► Adherence was assessed at each follow-up visit as good, fair and poor if a child missed <5%, <10% and >10% doses, respectively, of the expected monthly doses.

### Data analysis
The retrieved data were exported to IBM SPSS V.26.0 and STATA V.12.0 (STATA Corporation, College Station, Texas, USA), where they were processed for analysis. Descriptive statistics for categorical variables were analysed using the $X^2$ or Fisher's exact test and summarised using frequency and percentage. Depending on the data distribution, quantitative data were summarised using mean (±SD) or median (IQR). Suitable parametric or non-parametric statistics were used to evaluate differences. Incidence of attrition was calculated per 100 person-years (PY) (95% CI). Furthermore, Kaplan-Meier curves describe and compare retention and survival at different follow-up intervals by sex, address, cohort year, adjusted HR (AHR), initial cART backbone and record of cART change. The log-rank test was used to evaluate differences in the equality of survival functions. All LTFUs and transfers were censored on the date of their last visit. Estimates of HRs for attrition, 95% CI and quantification of associations were evaluated using Cox proportional hazards models. Log–log plots and plots of Schoenfeld residuals were used to evaluate the proportional hazards assumption. To adjust for possible confounders, multivariate Cox regression models were used. The final results are presented as adjusted hazard ratio (AHR) with 95% CI. All p values were two tailed and p values of <0.05 were considered statistically significant.

### Patient and public involvement
Patients or the public were not involved in the design, conduct, reporting or dissemination plans of our research.

## RESULTS
From September 2005 to December 2020, 822 CLHIV were enrolled in the Orotta National Paediatric Referral Hospital (ONPRH) paediatric cART treatment and follow-up programme. After evaluating eligibility, 710 (86.3%) participants were included in the analysis. Stratification of enrolment with respect to specific time frames showed that 480 (67.6%), 180 (25.4%), and 50 (7.0%) patients were enrolled to care between 2005–2010, 2011–2015, and 2016–2020, respectively (table 1). Eligible participants were followed up for 5364 PY follow-up. During this period, 65 (9.2%) died, 107 (15.1%) were LTFU, 153 (21.5%) were transferred out and 385 (54.2%) were currently receiving treatment.

### Sociodemographic characteristics of CLHIV
The median (IQR) age at enrolment in the clinic was 6 years (IQR: 3–9 years). The median (IQR) ages from 2005 to 2010; 2010–2015, and 2016–2020 were 5 years (IQR: 3–8 years), 7 years (IQR: 4–10 years), and 5 years (IQR: 6.25–12.75 years), respectively. In terms of sex, the numbers of boys and girls were similar (374 (52.7%) vs 336 (47.3%), respectively). Most of the participants were from the Maekel zone (72.7%).

### Clinical and disease-related characteristics of CLHIV at baseline
Assessment of clinic-level factors showed that the median CD4 count and percentage at treatment initiation were 274 cells/µL (IQR: 150–442 cells/µL) and 12% (7.4%–17%), respectively. In the relevant time frames, the median (IQR) values were 265 (151–392.7) cells/µL in 2005–2010; 307 (137–555) cells/µL in 2010–2015; and 340 (200–888) cells/µL in 2016–2020 (Mann-Whitney test, p=0.012). Similarly, the median (IQR) for CD4 percentages in 2005–2010, 2010–2015, and 2016–2020 were 11.1% (7.1%–16.5%), 13.6% (9%–21%), and 14.3% (7.521.9%), respectively (Mann-Whitney test, p=0.007). Overall, and in the following order, 519 (73.1%), 264 (45.7%), and 16 (2.4%) patients had advanced (stage III or IV) disease, anaemia, and TB before initiating cART, respectively. The proportion of children with WAZ <−2 SD (underweight), HAZ <−2 SD (stunted), and BAZ <−2 SD (undernourished) were 302 (68.1%), 461 (67.3%), and 247 (29.6%), respectively. Despite differences in enrolment in the successive time frames, no improvements were observed in these growth parameters (anaemia, WAZ, HAZ and BAZ) (table 1).

### Treatment-related characteristics and outcomes
The preferred cART backbone was zidovudine+lamivudine (AZT+3TC) (71.7%). Suboptimal adherence was recorded in 12.3% of patients, while most (72%) participants had a history of cART change at least once.

### Prevalence of retention and attrition
Overall attrition (death+LTFU), death and LTFU are presented in figure 2. Based on the results, the overall death rate and LTFU were 2.7 (95% CI: 2.1 to 3.5) per 100 PY and 1.2 (95% CI: 1 to 1.5) per 100 PY. Attrition rates at 0–≤6 months, >6–≤12 months, >12–≤24 months, >24–≤36 months, >36–≤48 months, and >48–≤60 months of follow-up were 3.6%, 2%, 2.9%, 1.3%, 1.4%, and 1.2%, respectively.

### Factors associated with retention and attrition
Out of 710 CLHIV included in the study, attrition occurred in 172 (24.2% (95% CI: 20% to 26.3%)) patients. The crude incidence and overall survival duration were 3.2 events per 100 PY (95% CI: 2.7 to 3.7) and 14 years (95% CI: 13.4 to 14.8). Duration of follow-up in years for attrition and retention was 4.5 years (IQR: 1–9 years) and 9 years (IQR: 6–11 years), respectively. In terms of proportions, attrition rates were higher in boys

**Table 1** Baseline characteristics of HIV-infected children and adolescents in the Orotta National Paediatric Referral Hospital (ONPRH) cART treatment centre, Asmara, Eritrea (2005–2020)

| Patients' characteristics | Total N (%) | 2005–2010 | 2011–2015 | 2016–2020 | P value (χ2) |
|---|---|---|---|---|---|
| Number enrolled | 710 | 480 (67.6) | 180 (25.4) | 50 (7.0) | |
| Gender | | | | | |
| Male | 374 (52.7) | 237 (63.4) | 111 (29.7) | 26 (7) | **0.01 (7.9)** |
| Female | 336 (47.3) | 243 (72.3) | 69 (20.5) | 24 (7.2) | |
| Age on enrolment in years, median (IQR) | 6 (3–9) | 5 (3–8) | 7 (4–10) | 5 (2–11) | **0.001*** |
| ≤5 | 274 (38.6) | 198 (72.3) | 52 (19) | 24 (8.7) | **<0.001 (27.5)** |
| 6–10 | 284 (40) | 202 (71.1) | 72 (25.4) | 10 (3.5) | |
| >10 | 152 (21.4) | 80 (52.6) | 56 (36.8) | 16 (10.6) | |
| Residence | | | | | |
| Maekel | 516 (72.7) | 358 (69.4) | 124 (24) | 34 (6.6) | 0.25 (2.7) |
| Outside Maekel | 194 (27.3) | 122 (62.9) | 56 (28.9) | 16 (8.2) | |
| Disease stage | | | | | |
| Early | 191 (26.9) | 134 (70.2) | 46 (24) | 11 (5.8) | 0.5 (1) |
| Advanced | 519 (73.1) | 346 (66.7) | 134 (25.8) | 39 (7.5) | |
| TB status | | | | | |
| Symptomatic and under treatment | 16 (2.4) | 7 (43.8) | 6 (37.5) | 3 (18.8) | 0.5 (5.9) |
| Not symptomatic | 625 (97.6) | 443 (70.9) | 135 (21.6) | 47 (7.5) | |
| CD4 count, median (IQR) | 274 (150–442) | 265 (151–392.7) | 307 (137–555) | 340 (200–888) | **0.012*** |
| CD4 percentage, median (IQR) | 12 (7.4–17) | 11.1 (7.1–16.5 | 13.6 (9–21) | 14.3 (7.5–21.9) | **0.007*** |
| Haemoglobin in g/L, mean (±SD) | 118 (±135) | 125 (±172) | 106 (±21) | 110 (±16) | **0.060*** |
| Anaemia | 264 (45.7) | 155 (58.7) | 89 (33.7) | 20 (7.6) | 0.3 (2.2) |
| No anaemia | 313 (54.3) | 197 (62.9) | 88 (28.2) | 28 (8.9) | |
| WAZ, median (IQR) | –2.6 (–3.6 to –1.7) | –2.5 (–3.5 to –1.7) | –2.8 (–3.8 to –1.7) | –2.9 (–3.8 to –1.6) | 0.263* |
| WAZ ≥–2 | 141 (31.9) | 98 (69.5) | 32 (22.7) | 11 (7.8) | 0.7 (0.5) |
| WAZ <–2 | 302 (68.1) | 202 (66.9) | 78 (25.8) | 22 (7.3) | |
| HAZ, median (IQR) | –2.75 (–3.8 to –1.7) | –2.6 (–3.7 to –1.5) | –2.9 (–4 to –1.8) | –3 (–3.9 to –1.8) | 0.063* |
| HAZ ≥–2 | 223 (32.7) | 161 (72.2) | 49 (22) | 13 (5.8) | 0.1 (4.3) |
| HAZ <–2 | 461 (67.3) | 296 (64.2) | 129 (28) | 36 (7.8) | |
| BAZ, median (IQR) | –1.58 (–2.8 to –0.69) | –1.5 (–2.9 to –0.5) | –1.6 (–2.7 to –1) | –1.7 (–3 to –0.6) | 0.385* |
| BAZ ≥–2 | 378 (60.4) | 257 (68) | 97 (25.7) | 24 (6.3) | |
| BAZ <–2 | 247 (29.6) | 162 (65.6) | 64 (25.9) | 21 (8.5) | 0.5 (1) |
| cART backbone | | | | | |
| AZT+3TC | 508 (71.7) | 347 (68.3) | 148 (29.1) | 13 (2.6) | |
| Other backbones | 201 (28.3) | 133 (66.2) | 31 (15.4) | 37 (18.4) | **<0.001 (62)** |
| NNRTI | | | | | |
| Efavierenz (EFV) | 366 (51.6) | 262 (71.6) | 90 (24.6) | 14 (3.8) | **<0.001 (13)** |
| Nevirapin (NVP) | 340 (48.4) | 216 (63.5) | 88 (25.8) | 36 (10.6) | |
| Current cART regimen | | | | | |
| First-line cART | 633 (89.2) | 429 (67.8) | 157 (24.8) | 47 (7.4) | 0.3 (1.9) |
| Second-line cART | 77 (10.8) | 51 (66.2) | 23 (29.9) | 3 (3.9) | |

Continued

**Table 1** Continued

| Patients' characteristics | Total N (%) | 2005–2010 | 2011–2015 | 2016–2020 | P value ($\chi$2) |
|---|---|---|---|---|---|
| Adherence | | | | | |
| Suboptimal adherence | 91 (12.3) | 46 (50.5) | 39 (42.9) | 6 (6.6) | **<0.001 (17)** |
| No record | 619 (87.2) | 434 (70) | 141 (22.8) | 44 (7.2) | |
| cART change | | | | | |
| cART changed at least once | 516 (75) | **340 (65.9)** | 148 (28.7) | 28 (5.4) | **<0.001 (1.2)** |
| No cART change | 172 (25) | 120 (69.8) | 30 (17.4) | 22 (12.8) | |
| Duration of follow-up in years, median (IQR) | 8 (4–11) | 10 (7–12) | 6 (4–8) | 2 (1–3.25) | **<0.001*** |

Comparisons of proportions were performed by using the $\chi$2 test, medians by using Mann-Whitney tests.
Other backbones refer to tenofovir disoproxil fumarate+emtricitabine, abacavir+3TC or stavudine+3TC. Anaemia was defined as haemoglobin level <110 g/L for children <5 years old, <115 g/L for children 5–11.9 years old and <120 g/L for children >12 years old.
*Mann-Whitney U test.
AZT+3TC, zidovudine+lamivudine; BAZ, body mass index-for-age z-score; cART, combined antiretroviral therapy; EFV, Efavirenz; HAZ, height-for-age z-score; IQR, Inter-quartile range; NNRTI, non-nucleoside reverse transcriptase inhibitor; NVP, Nevirapin; TB, tuberculosis; WAZ, weight-for-age z-score.

104 (27.8%) vs 68 (20.2%) in girls; residence outside Maekel 57 (29.4%) vs 115 (22.3%) in residence within Maekel; advanced HIV (WHO stage III and IV) 149 (28.7) vs 23 (12%) for early presentation; presence of anaemia 72 (27.5%) vs 54 (17.3%) in no anaemia; WAZ <−2 (underweight) 83 (27.5%) vs 27 (19.1%) in WAZ >−2; HAZ <−2 (stunting) 122 (26.5%) vs 42 (18.8%) in HAZ >−2; BAZ <−2 (undernourished) 73 (29.6%) vs 76 (20.1%) in BAZ >−2; cART other than AZT+3TC backbone 71 (35.3%) vs 101 (19.9%) in AZT+3TC; no record of cART change or switching 77 (44.8%) vs 92 (17.8%) (table 2).

## Kaplan-Meier analysis for attrition incidence

Kaplan-Meier survival curves with log-rank tests were constructed to compare the cumulative incidence of attrition by sex, address, cohort year, AHR, initial cART backbone and record of cART change (figure 3 and table 3). Overall, boys had a shorter survival of 13 years (95% CI: 2 to 14 years) vs 13.8 years in girls (95% CI: 13 to 14 years) (log-rank test, p=0.008) (figure 3B). The mean survival duration was significantly shorter with increasing age at baseline: ≤5 years: 15.1 (14.2–16) years vs 6–10 years: 11.7 (11.1–12.2 years) and >10 years: 8.4 (7.43–12.2 years) years (log-rank test, p<0.001). Children

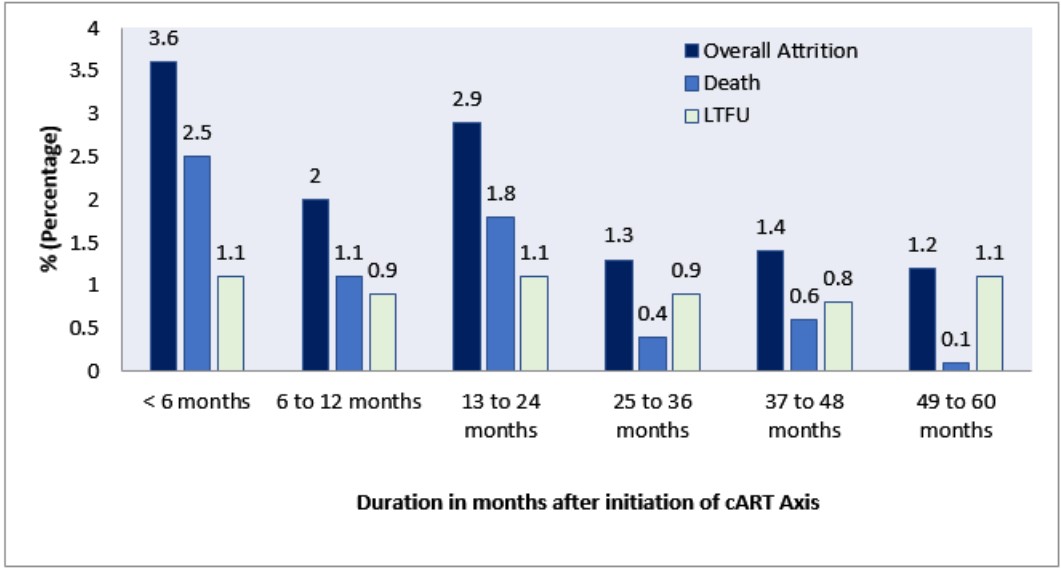

**Figure 2** Frequency of attrition, deaths and LTFU at specified time points within the first 60 months after initiation of cART. cART, combined antiretroviral therapy; LTFU, lost to follow-up.

Table 2 Characteristics of the study participants stratified by survival outcome in the Orotta National Paediatric Referral Hospital (ONPRH) cART treatment centre, Asmara, Eritrea (2005–2020)

| Cohort characteristics | Total n (%) | Attrition n (%) | Retention n (%) | P value ($\chi2$) |
|---|---|---|---|---|
| Gender | | | | |
| Male | 374 (52.7) | 104 (27.8) | 270 (72.2) | **0.01 (5.5)** |
| Female | 336 (47.3) | 68 (20.2) | 268 (79.8) | |
| Age on enrolment in years, median (IQR) | 6 (3–9) | 6 (3–10) | 6 (3–9) | 0.38* |
| ≤5 | 274 (38.6) | 62 (22.6) | 212 (77.4) | 0.09 (4.7) |
| 6–10 | 284 (40) | 63 (22.2) | 221 (77.8) | |
| >10 | 152 (21.4) | 47 (30.9) | 105 (69.1) | |
| Enrolment year, median (IQR) | | | | 0.24* |
| 2005–2010 | 265 (37.3) | 111 (23.1) | 369 (76.9) | 0.23 (2.9) |
| 2011–2015 | 289 (40.7) | 44 (24.4) | 136 (75.6) | |
| 2016–2020 | 156 (22) | 17 (34) | 33 (66) | |
| Residence | | | | |
| Maekel | 516 (72.7) | 115 (22.3) | 401 (77.7) | **0.049 (3.8)** |
| Outside Maekel | 194 (27.3) | 57 (29.4) | 137 (70.6) | |
| Disease stage | | | | |
| Early | 191 (26.9) | 23 (12) | 168 (88) | **<0.001 (21)** |
| Advanced | 519 (73.1) | 149 (28.7) | 370 (71.3) | |
| TB status | | | | |
| Symptomatic and under treatment | 16 (2.4) | 5 (31.2) | 11 (68.8) | 0.5 (0.4) |
| Not symptomatic | 625 (97.6) | 151 (24.2) | 474 (75.8) | |
| CD4 count, median (IQR) | 274 (150–442) | 216 (100–371) | 287 (163–452) | **<0.001 *** |
| CD4 percentage, median (IQR) | 12 (7.4–17) | 9.9 (6–14.5) | 12.5 (8.2–19) | **<0.001*** |
| Haemoglobin in g/L, mean (±SD) | 118 (±135) | 107 (±19) | 122(±152) | **<0.001†** |
| Anaemia | 264 (45.7) | 72 (27.5) | 192 (72.5) | **0.004 (8.4)** |
| No anaemia | 313 (54.3) | 54 (17.3) | 259 (82.7) | |
| WAZ, median (IQR) | −2.6 (−3.6 to −1.7) | −3 (−4.3 to −2) | −2.5 (−3.5 to −1.6) | **0.001*** |
| WAZ ≥−2 | 141 (31.9) | 27 (19.1) | 114 (80.9) | **0.05 (3.5)** |
| WAZ <−2 | 302 (68.1) | 83 (27.5) | 219 (72.5) | |
| HAZ, median (IQR) | −2.75 (−3.8 to −1.7) | −3.2 (−4 to −1.9) | −2.5 (3.7 to −1.5) | **0.001*** |
| HAZ ≥−2 | 223 (32.7) | 42 (18.8) | 181 (81.2) | **0.028 (4.8)** |
| HAZ <−2 | 461 (67.3) | 122 (26.5) | 339 (73.5) | |
| BAZ, median (IQR) | −1.58 (−2.8 to −0.69) | −1.9 (−3.6 to −0.7) | −1.5 (−2.6 to −0.6) | **0.012*** |
| BAZ ≥−2 | 378 (60.4) | 76 (20.1) | 302 (79.9) | **0.007 (7.3)** |
| BAZ <−2 | 247 (29.6) | 73 (29.6) | 174 (70.4) | |
| cART backbone | | | | |
| AZT+3TC | 508 (71.7) | 101 (19.9) | 407 (80.1) | **<0.001 (18.6)** |
| Other backbones | 201 (28.3) | 71 (35.3) | 130 (64.7) | |
| NNRTI | | | | |
| Efavirenz (EFV) | 366 (51.6) | 91 (24.9) | 275 (75.1) | 0.67 (0.1) |
| Nevirapin (NVP) | 340 (48.4) | 80 (23.5) | 260 (76.5) | |
| Current cART regimen | | | | |
| First-line cART | 633 (89.2) | 156 (24.6) | 477 (75.4) | 0.45 (0.5) |
| Second-line cART | 77 (10.8) | 16 (20.8) | 61 (79.2) | |

Continued

**Table 2** Continued

| Cohort characteristics | Total n (%) | Attrition n (%) | Retention n (%) | P value (χ2) |
|---|---|---|---|---|
| Adherence | | | | |
| Suboptimal adherence | 91 (12.3) | 24 (26.4) | 67 (73.6) | 0.6 (0.2) |
| No record | 619 (87.2) | 148 (23.9) | 471 (76.1) | |
| cART change | 516 (75) | 92 (17.8) | 424 (82.2) | **<0.001 (50.5)** |
| No cART change | 172 (25) | 77 (44.8) | 95 (55.2) | |
| Duration of follow-up in years, median (IQR) | 8 (4–11) | 4.5 (1–9) | 9 (6–11) | **<0.001*** |

Comparisons of proportions were performed by using the χ2 test, medians by using Mann-Whitney tests and means using independent sample t-test.

Other backbones refer to tenofovir disoproxil fumarate+emtricitabine, abacavir+3TC or stavudine+3TC. Anaemia was defined as haemoglobin level <11 g/dL for children <5 years old, <11.5 g/dL for children 5–11.9 years old and <12 g/dL for children >12 years old.

*Mann-Whitney U test.

†Independent sample t-test.

AZT+3TC, zidovudine+lamivudine; BAZ, body mass index-for-age z-score; cART, combined antiretroviral therapy; EFV, Efavirenz; HAZ, height-for-age z-score; NNRTI, non-nucleoside reverse transcriptase inhibitor; NVP, Nevirapin; TB, tuberculosis; WAZ, weight-for-age z-score.

who reside outside Maekel also have a significantly shorter mean survival of 11 years (95% CI: 10 to 11.7 years) in the clinic compared with those from Maekel with 14.5 years (95% CI: 13.8 to 15.4 years) (log-rank test, p=0.002 (figure 3C)). Late enrolment was also associated with shorter mean survival (2005–2009: 14.8 years (95% CI: 14 to 15.5 years) vs 2010–2014: 7.5 years (95% CI: 7 to 8 years) vs 2015–2020: 3.1 years (95% CI: 2.7 to 3.5 years) (log-rank test, p=0.008) . Children with advanced HIV disease at presentation had a significantly shorter mean survival time in ONPRH (13 years (95% CI: 12.3 to 14)) compared with those without (15 years (95% CI: 14.2 to

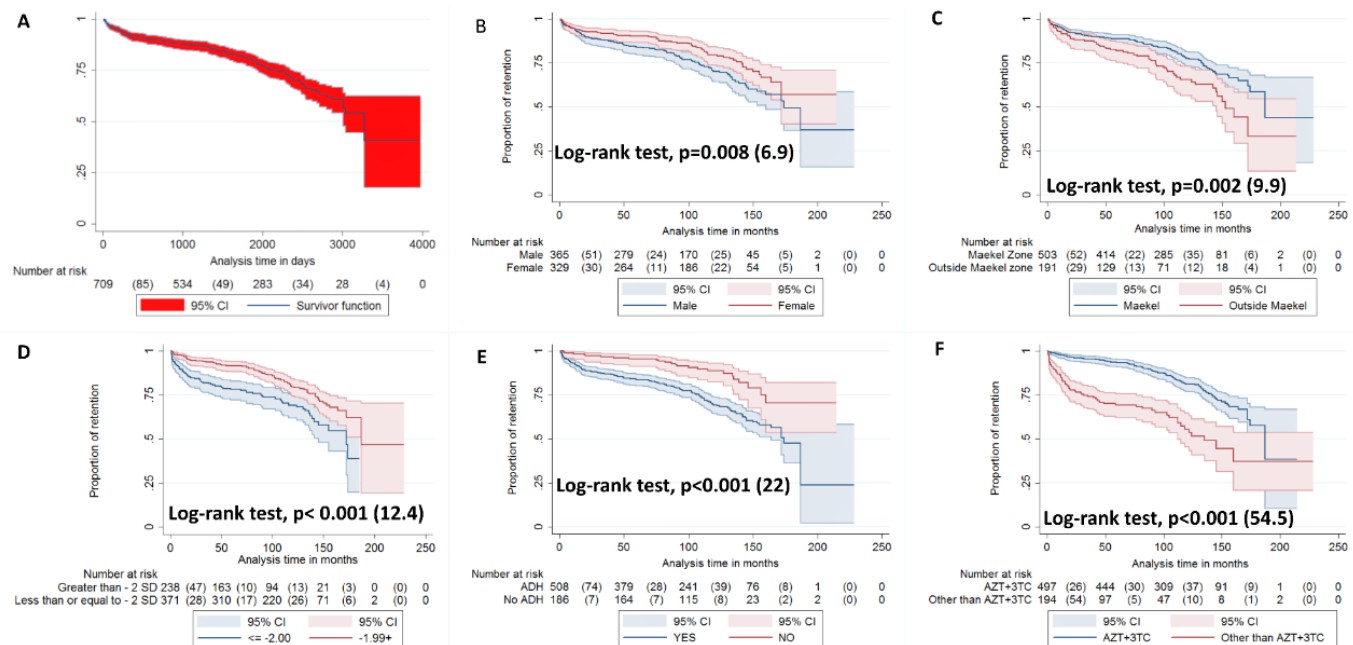

**Figure 3** Kaplan-Meier cumulative incidence of attrition unadjusted curves for children followed in Orotta National Paediatric Referral Hospital (ONPRH) from 2005 to 2020. Figure 3A: Overall rate of retention for the entire cohort, Figure 3B: Gender wise comparison of retention rates, Figure 3C: Address stratification of retention rates, Figure 3D: Retention rate per BMI for age, greater than or less than -2 SD, Figure 3E: Retention rate among early versus advanced clinical stage, Figure 3F: cART backbone categories and retention rate

**Figure Footnote: General : Other backbones refer to tenofovir disoproxil fumarate+emtricitabine, abacavir+3TC or stavudine+3TC.**

**Abbreviations: AZT+3TC: zidovudine+lamivudine and AHD: Advanced HIV Disease**

**Table 3** Crude incidence, rate of attrition and Kaplan-Meier survival estimates of children followed in the Orotta National Paediatric Referal Hospital (ONPRH) cART treatment centre, Asmara, Eritrea (2005–2020)

| Cohort characteristics | Incidence of attrition per 100 person years (95% CI) | Mean survival duration in years (95% CI) | P value (log-rank) |
|---|---|---|---|
| Total | 3.2 (2.7 to 3.7) | 14.1 (13.4 to 14.8) | |
| Gender | | | |
| Male | 3.8 (31. to 4.6) | 13.2 (2.1 to 14.3) | **0.008 (6.9)** |
| Female | 2.5 (2 to 3.2) | 13.8 (13.1 to 14.4) | |
| Age in years at enrolment | | | |
| ≤5 | 2.5 (1.9 to 3.2) | 15.1 (14.2 to 16) | **<0.001 (36.8)** |
| 6–10 | 2.8 (2.2 to 3.6) | 11.7 (11.1 to 12.2) | |
| >10 | 6.7 (5 to 8.9) | 8.4 (7.43 to 9) | |
| Residence | | | **0.002 (9.8)** |
| Maekel | 2.7 (2.3 to 3.3) | 14.6 (13.8 to 15.4) | |
| Outside Maekel | 4.6 (3.5 to 5.9) | 10.9 (10 to 11.7) | |
| Disease stage at presentation | | | **<0.001 (22)** |
| Early | 1.4 (0.9 to 2.1) | 15 (14.2 to 15.7) | |
| Advanced | 3.9 (3.3 to 4.6) | 13.2 (12.3 to 14) | |
| TB status | | | 0.2 (1.5) |
| Symptomatic and under treatment | 3.1 (2.6 to 3.7) | 9.9 (6.9 to 12.8) | |
| Not symptomatic | 6.5 (2.7 to 15.8) | 14 (13.3 to 14.8) | |
| Haemoglobin | | | **0.004 (8.3)** |
| Anaemia | 2.2 (1.7 to 2.9) | 11.5 (11 to 12) | |
| Normal | 3.7 (2.9 to 4.7) | 12.8 (12 to 13) | |
| WAZ | | | **0.014 (6)** |
| WAZ ≥−2 | 2.1 (1.48 to 3.1) | 13 (12.4 to 13.7) | |
| WAZ <−2 | 3.5 (2.8 to 4.4) | 12 (11.4 to 12.8) | |
| HAZ | | | **0.019 (5.4)** |
| HAZ ≥−2 | 2.3 (1.7 to 3.2) | 14.9 (13.7 to 16) | |
| HAZ <−2 | 3.5 (3 to 4.2) | 12.7 (12 to 13.4) | |
| BAZ | | | **<0.001 (12.4)** |
| BAZ ≥−2 | 2.4 (1.9 to 3) | 15 (14 to 15.9) | |
| BAZ <−2 | 4.4 (3.5 to 5.5) | 12 (11 to 13) | |
| Initial cART backbone | | | **<0.001 (54.5)** |
| AZT+3TC | 2.3 (1.9 to 2.8) | 13.9 (13.4 to 14.5) | |
| Other backbones | 7.1 (5.6 to 9) | 11.3 (9.8 to 12.8) | |
| NNRTI | | | 0.9 (0.015) |
| Efavirenz (EFV) | 3.2 (2.6 to 3.9) | 13 (12.3 to 13.7) | |
| Nevirapin (NVP) | 3 (2.5 to 3.9) | 14.2 (13.3 to 15.2) | |
| Current cART regimen | | | 0.15 (2) |
| First-line cART | 3.34 (2.8 to 3.9) | 12.9 (12.3 to 13.4) | |
| Second-line cART | 2.2 (1.3 to 3.7) | 15.7 (14.3 to 17) | |
| Adherence | | | 0.68 (0.16) |
| Suboptimal adherence | 2.9 (1.9 to 4.3) | 12.6 (11.8 to 13.4) | |
| No record | 3.2 (2.7 to 3.7) | 14.3 (13.6 to 15) | |

**Table 3** Continued

| Cohort characteristics | Incidence of attrition per 100 person years (95% CI) | Mean survival duration in years (95% CI) | P value (log-rank) |
|---|---|---|---|
| cART change | | | <0.001 (136) |
| cART change | 1 (0.84 to 1.3) | 13.6 (13 to 14) | |
| No cART change | 2 (1.7 to 2.5) | 7.5 (6.6 to 8.4) | |

Comparisons of survival duration were performed by using the log-rank test and its p value.
Other backbones refer to tenofovir disoproxil fumarate+emtricitabine, abacavir+3TC or stavudine+3TC. Anaemia was defined as haemoglobin level <11 g/dL for children <5 years old, <11.5 g/dL for children 5–11.9 years old and <12 g/dL for children >12 years old.
AZT+3TC, zidovudine+lamivudine; BAZ, body mass index-for-age z-score; cART, combined antiretroviral therapy; HAZ, height-for-age z-score; NNRTI, non-nucleoside reverse transcriptase inhibitor; TB, tuberculosis; WAZ, weight-for-age z-score.

15.7 years)) (figure 3D). Wasted children (BAZ score <−2 SD (12 years (11–13 years)) showed lower retention than their counterpart (BAZ>-2 SD) (figure 3E). The survival time for those taking treatment other than AZT+3TC (11.3 years (95% CI: 9.8 to 12.8 years)) was significantly shorter compared with the AZT+3TC group (14 years (95% CI: 13.4 to 14.5 years)) (log-rank test, p<0.001) (figure 3F). Moreover, children with a record of cART change had significantly longer mean survival time in the clinic (13 years (95% CI: 13 to 14 years)) as compared with those without (7.5 years (95% CI: 6.6 to 8.4 years)) (log-rank test, p<0.001). Other factors associated with shorter survival were: anaemia at ART initiation (haemoglobin <8 g/dL) (p=0.032); WAZ <−2 (12 years (11.4–12.8 years)); HAZ <−2 (12.7 years (12–13.4 years))).

### Independent predictors of attrition rate in children enrolled in ONPRH cART treatment and follow-up clinic

The HRs (unadjusted and adjusted) for attrition in 701 patients and associated p values for differences in attrition across specific cohort characteristics are shown in table 4. In the univariate Cox proportional hazards model, increased risk of attrition was associated with the following: boys (unadjusted HR (UHR)=1.4, 95% CI: 1 to 2, p=0.01); residence outside Zoba Maekel (UHR=1.6, 95% CI: 1.2 to 2.2, p=0.002); WHO BAZ <−2 (UHR=1.8, 95% CI: 1.3 to 2.5, p=0.001); WHO HAZ (UHR=1.5, 95% CI: 1 to 2.1, p=0.02); presence of anaemia (UHR=1.7, 95% CI: 1.7 1.2 to 2.4, p=0.005); advanced disease (WHO III or IV) at presentation (UHR=2.9, 95% CI: 1.8 to 4.3, p<0.001); initiation of AZT+3TC or other cART backbones (UHR=3, 95% CI: 2.2 to 4.2, p<0.001). In contrast, reduced risk of attrition was associated with cART changes (UHR=0.16, 95% CI: 0.12 to 0.2, p<0.001).

In the adjusted Cox proportional hazards model, independent predictors of attrition were male sex (AHR=1.6, 95% CI: 1 to 2.4, p=0.01); residence outside Zoba Maekel (AHR=1.5, 95% CI: 1 to 2.3, p=0.02); later enrolment years (2010–2015: AHR=3.2, 95% CI: 1.9 to 5.3, p<0.001; >2015: UHR=6.1, 95% CI: 3 to 12.2, p<0.001); BAZ <−2 (AHR=1.4, 95% CI: 0.9 to 2.1, p=0.05); advanced HIV disease (WHO III or IV) at enrolment (AHR=2.2, 95% CI: 1.2 to 3.9, p=0.004); initiation of AZT+3TC or other cART backbones (UHR=2, 95% CI: 1.2 to 3.2, p=0.005). In contrast, a reduced likelihood of attrition was observed in children with a record of cART changes (UHR=0.2, 95% CI: 0.15 to 0.4, p<0.001) (table 4).

### DISCUSSION

Attrition rates among those receiving care were lower than those reported in similar studies in SSA.[6] However, our finding of high attrition and mortality rates in the first 6 months of treatment is consistent with multiple reports.[5 13 14] In the multicounty treated Asia Paediatric HIV Observational Database, mortality decreased from 10.2 (95% CI: 7.5 to 13.7) deaths/100 PY in the first 3 months to 4.2 (95% CI: 2.6 to 6.8) deaths/100 PY in the fourth and sixth months.[15] The high mortality in the first 6 months of enrolment in care has largely been attributed to late presentation (WHO stage III or IV), chronic diarrhoea and weight-for-height z-score <−2 (severe wasting). Observational studies suggest that the first 6 months after treatment initiation is the most vulnerable period for HIV-positive children in SSA.[5] Studies consistently cite the need for context-specific mitigation (short messaging service-based messaging/reminder; improved linkage of child and maternal HIV/AIDS services; intensive case management),[5] directed at maximising retention at this stage.

Furthermore, analysis of attrition incidence demonstrated a rate of 3.2 (95% CI: 2.7 to 3.7) per 100 PY. This rate estimate is lower compared with studies conducted in Ethiopia (8.36 (95% CI: 7.12 to 9.80) per 100 PY),[16] Zimbabwe (11.8 (95% CI: 11.0 to 12.7) per 100 PY),[17] Nigeria (10.8/100 PY)[18] and Kenya (23.1/100 PY).[19] In contrast, the incidence of attrition in this study was higher compared with other studies.[14 15 20] Variation in incidence between countries in SSA is caused by differences in access to and quality of healthcare services, health-seeking behaviours between populations,[21] heterogeneity in methodology and definition of outcomes. Finally, it is possible that the LTFUs registered in our study may be due to undocumented mortality instead of disengagement

**Table 4** Cox proportional hazards of attrition among CLHIV and adolescents at Orotta National Paediatric Referral Hospital (ONPRH) cART treatment centre, Asmara, Eritrea (2005–2020)

| Cohort characteristics | Unadjusted HR (95% CI) | P value | Adjusted HR (95% CI) | P value |
|---|---|---|---|---|
| Gender | | | | |
| Female | 1 (ref) | 0.01 | 1 (ref) | 0.01 |
| Male | 1.4 (1 to 2) | | 1.6 (1 to 2.4) | |
| Age at enrolment | | | | |
| ≤5 | 1 (ref) | | 1 (ref) | |
| 6–10 | 1.3 (0.9 to 1.8) | | 1.1 (0.7 to 1.6) | |
| >10 | 3.3 (2.2 to 5) | | 1.5 (0.9 to 2.5) | |
| Residence | | | | |
| Maekel | 1 (ref) | 0.002 | 1 (ref) | 0.02 |
| Outside Maekel | 1.6 (1.2 to 2.2) | | 1.5 (1 to 2.3) | |
| Year of enrolment | | | | |
| <2010 | 1 (ref) | | 1 (ref) | |
| 2010–2015 | 1.9 (1.3 to 2.7) | | 3.2 (1.9 to 5.3) | <0.001 |
| >2015 | 4.2 (2.5 to 7.2) | | 6.1 (3 to 12.2) | <0.001 |
| BMI-for-age z-score | | | | |
| ≥2 | 1 (ref) | 0.001 | 1 (ref) | 0.03 |
| <−2 | 1.8 (1.3 to 2.5) | | 1.4 (0.9 to 2.1) | |
| Height-for-age z-score | | | | |
| ≥2 | 1 (ref) | 0.02 | | |
| <−2 | 1.5 (1 to 2.1) | | | |
| Haemoglobin | | | | |
| Normal | 1 (ref) | 0.005 | | |
| Anaemia | 1.7 (1.2 to 2.4) | | | |
| Advanced disease at presentation | | | | |
| No | 1 (ref) | <0.001 | 1 (ref) | 0.004 |
| Yes | 2.9 (1.8 to 4.3) | | 2.2 (1.2 to 3.9) | |
| TB status | | | | |
| Not symptomatic | 1 (ref) | 0.23 | | |
| Took anti-TB | 1.8 (0.8 to 5) | | | |
| cART backbone | | | | |
| AZT+3TC | 1 (ref) | <0.001 | 1 (ref) | 0.005 |
| Other backbones | 3 (2.2 to 4.2) | | 2 (1.2 to 3.2) | |
| NNRTI | | | | |
| Efavirenz (EFV) | 1 (ref) | 0.9 | | |
| Nevirapin (NVP) | 0.95 (0.7 to 1.2) | | | |
| cART changes | | | | |
| No | 1 (ref) | <0.001 | 1 (ref) | <0.001 |
| Yes | 0.16 (0.12 to 0.2) | | 0.2 (0.15 to 0.4) | |
| Current cART regimen | | | | |
| First-line cART | 1 (ref) | 0.16 | | |
| Second-line cART | 0.68 (0.4 to 1.1) | | | |
| Suboptimal adherence | | | | |
| No | 1 (ref) | 0.2 | | |
| Yes | 1.1 (0.7 to 1.7) | | | |

Other backbones refer to tenofovir disoproxil fumarate+emtricitabine, abacavir+3TC or stavudine+3TC. Anaemia was defined as haemoglobin level <11 g/dL for children <5 years old, <11.5 g/dL for children 5–11.9 years old and <12 g/dL for children >12 years old.
AZT+3TC, zidovudine+lamivudine; BMI, body mass index; cART, combined antiretroviral therapy; CLHIV, children living with HIV; NNRTI, non-nucleoside reverse transcriptase inhibitor; TB, tuberculosis.

from care or undocumented transfers. Therefore, the mortality rate may have been underestimated.

In many aspects, the predictors of attrition identified in this study are largely similar to other regional studies. For instance, evidence from multiple studies in the region indicates that WHO BAZ <−2 (undernutrition) and BAZ <−2 are baseline hazards for cART attrition.[8 10 17] Although the presence of anaemia at baseline was associated with an increased risk of attrition (1.7; 95% CI: 1.2 to 2.4), the association was attenuated in the adjusted multivariable model. This suggests the presence of confounders. Nevertheless, the conjunction of these parameters should not be ignored. Several reports suggest that severe malnutrition (SAM) and HIV are associated with increased morbidity and mortality in affected infants.[22 23] Others have argued that HIV+SAM represents a distinct disease stratum/phenotype with a worse prognosis and impaired immune recovery.[24] Importantly, managing these patients is extremely challenging.[12] Highlighting this point, some investigators suggest that rapid initiation (within 1 week) of cART in children with SAM or WHO stage III or IV disease may be counterproductive, and that treatment in these patients should be delayed until the patient is stabilised.[23] Based on these considerations, the observed connection between advanced HIV disease (III or IV) at enrolment and attrition rate (AHR=2.2, 95% CI: 1.2 to 3.9) was predictable.[13] Reports from Zimbabwe,[17] Nigeria[18] and Myanmar[25] have demonstrated that late presentation remains a challenge in this region.

Additional factors unique to this setting that were associated with attrition included male sex, residence outside Asmara, enrolment in later years and late presentation. In general, data on the relationship between sex and attrition are inconsistent, with most studies reporting a null relationship, while others skew toward either sex.[8 26 27] We believe that these inconsistencies and paradoxical results in cohorts across SSA mirror the disparate context-specific influences in the region. In contrast, the observed relationship between attrition and residence outside Asmara has greater practical implications. First, experts mention that HIV-related discrimination and internalised shame may constrain access to healthcare among adults living with HIV (ALWH).[28] For context, ridicule from friends/peers may force guardians/parents to enrol patients in distant facilities outside Zoba Maekel. This practice can compromise linkage and retention in care by imposing unnecessary out-of-pocket costs (transport).[29–31] Fear of victimisation can also prompt delays or failure by parents/guardians to disclose perinatally acquired HIV status,[32] a phenomenon which can promote late presentation. In many ways, the highlighted connection between stigma, attrition, enrolment in later years and late presentation is a common finding in attrition studies on ALWH.[13 14 22] We concede that HIV programmes in Eritrea are ill-equipped to deal with the complex needs of ALWH. Multiple studies demonstrated that ALWH recruitment and retention in care are undermined by a lack of psychosocial support, lack of health personnel trained on ALWH-related issues, barriers to care uptake and limited support during the transition to adult care.[33] Implementing comprehensive interventions targeting these barriers will require robust research.

Lastly, in the multivariable proportional hazards regression, the adjusted hazards of attrition were highest in children without cART changes or on other cART backbones (tenofovir disoproxil fumarate+emtricitabine, abacavir+3TC or stavudine+3TC). A causal analysis of these associations revealed some subtle connections. Retention in care may correlate with a higher likelihood of cART changes/switching to other backgrounds. In contrast, late presentation is associated with a higher likelihood of mortality/attrition within the first 6 months of enrolment.[34] This can undercut the prospects of cART switches, particularly in settings with a limited range of cART options. If valid, then the latter explanation highlights a major systematic weakness in cART clinics across Eritrea. There is limited diversity of cART options for CLHIV and ALWH.

### Strength and limitations of the study

A major strength of this study is the use of routinely collected information from the largest paediatric cART clinic in Eritrea. Therefore, the results can provide a useful gauge of real-world programme successes and weaknesses. However, our study has limitations. Retrospective designs suffer from multiple constraints, including missing data. Missing or unstructured collection of information (CD4$^+$ cell counts or viral load data) and lack of data on the causes of death and outcomes in children characterised as LTFU are particular examples. In the absence of tracing information, mortality estimates in this cohort are unclear.

**Acknowledgements** We are grateful to the National Communicable Disease Control Division, Eritrean Ministry of Health, and ART Health Management Information System (HMIS) developers and technicians. Our sincere appreciation goes to Dr Araia Berhane, Dr Fanus Yemane, Dr Simon Tesfay, Dr Yonathan Tesfalidet and Mr Thomas Asfaha.

**Contributors** All authors contributed to the conceptualisation of the study and approved the final draft.Conceptualisation—STM. Data curation—STM and GGG. Formal analysis—STM and OOA. Investigation—STM, MEH and OOA. Methodology—STM, GGG and AR. Project administration—STM, MMI, TGT, MEH and OOA. Writing (original draft)—STM and OOA. Writing (review and editing)—STM, GGG, OOA, MMI, TGT and MEH.

**Funding** The authors have not declared a specific grant for this research from any funding agency in the public, commercial or not-for-profit sectors.

**Competing interests** None declared.

**Patient and public involvement** Patients and/or the public were not involved in the design, or conduct, or reporting, or dissemination plans of this research.

**Patient consent for publication** Not required.

**Ethics approval** Ethical approval was obtained from the Ministry of Health (MOH) Research Ethics and Protocol Review Committee (letter of reference: 02/05/21). All data collected from the respondents' records were kept confidential. Consent to participate was not obtained from the patients or guardians because the study was based on anonymised patient records, which was waived by the ethical committee. All study procedures followed the recommendations of the Declaration of Helsinki Convention.

**Provenance and peer review** Not commissioned; externally peer reviewed.

**Data availability statement** Data are available upon reasonable request. The datasets supporting the conclusions of this article are available from the corresponding author on reasonable request.

**Open access** This is an open access article distributed in accordance with the Creative Commons Attribution 4.0 Unported (CC BY 4.0) license, which permits others to copy, redistribute, remix, transform and build upon this work for any purpose, provided the original work is properly cited, a link to the licence is given, and indication of whether changes were made. See: https://creativecommons.org/licenses/by/4.0/.

**ORCID iDs**
Samuel Tekle Mengistu http://orcid.org/0000-0002-2817-1421
Ghirmay Ghebrekidan Ghebremeskel http://orcid.org/0000-0002-0275-2279
Mohammed Elfatih Hamida http://orcid.org/0000-0002-5337-399X
Oliver Okoth Achila http://orcid.org/0000-0001-8013-0785

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
