## [Reviewer comments · BMJ Paediatrics Open]

ARTICLE DETAILS

TITLE (PROVISIONAL)	Attrition and associated factors among HIV-infected children at Orotta National Referral and Teaching Hospital, Asmara, Eritrea: A 15-year cohort analysis
AUTHORS	Mengistu, Samuel Ghebremeskel, Ghirmay Rezene, Aron Idris, Mahmud Mohammed Tikue, Tsegereda Gebrehiwot Hamida, Mohammed Elfatih Achila, Oliver Okoth

VERSION 1 – REVIEW

REVIEWER	Reviewer name: Dr. Rachel Mary Hilliam Institution and CountryThe Open University Mathematics and Statistics The Open University Walton Hall, United Kingdom of Great Britain and Northern Ireland Competing interests: None
REVIEW RETURNED	11-Feb-2022

GENERAL COMMENTS	This is an interesting paper which I'm sure will be of relevance to the readers of the journal. It represents a reasonably comprehensive longitudinal study. There are a few statistics issues which would benefit from clarification. The first is what exactly the p-values relate to in Table 1. Are these chi-square for total numbers across different years (ie male/female three year groups), in other words treating this as a 3x2 contingency table? If so then I assume what you are trying to look for here is that the proportions remain roughly the same across the 3 groups, as these are baseline characteristics. This is not entirely clear if this is the case in your results section which discusses the baseline characteristics as most of this section repeats what is given in the table rather than adding an explanation to the results. Similarly in Table 2 you need to be clear what these p-values relate to. In this case are you comparing number of Attrition with number of retention in each of the categories. I think the different years of attrition and retention also need some explanation, particularly in terms of what this means for the data. In Table 3 I'm assuming the p-values here from the the proportional hazards? This again isn't clear. In Table 4 and throughout the paper it is usual to write confidence intervals with the lower and upper limit in round brackets, not with - separating the two limits. So (LL,UL) would be standard statistical notation. I would also suggest that you need much more context around the results in Table 4 for the readers of this journal. I would suggest a slight restructuring and putting some of the
---

	context which is in the discussion into the appropriate subsections of the results. This would help readers understand exactly what is being analysed and put the results in context for each of the Tables. There are a number of small grammatical errors which should be picked up. Overall it is a paper which is of interest to the readers of the journal, but it needs more explanation of the analysis and results.
--	--

REVIEWER	Reviewer name: Prof. Lee Fairlie Institution and Country: Wits RHI Maternal and Child, South Africa Competing interests: None
REVIEW RETURNED	19-Mar-2022

GENERAL COMMENTS	Thank you for the opportunity to review this manuscript, it is interesting and generally very well written. I have listed suggestions regarding a major revision below:  - Suggest using children living with HIV (CLHIV) or HIV-positive children throughout and remaining consistent Abstract  - Line 24 "Among the 710....."seems to stop short and should be revised - The manuscript repeatedly refers to "residence outside Zoba Maekel" which seems to predict worse outcomes, however this is not very clear for people unfamiliar with this city/region so please elaborate on why this may be important-is it more rural/less expertise/fewer children and therefore less confidence etc. - Please spell out in full BAZ/WAZ/HAZ and other abbreviations the first time they're used. - Change to another regimen is cited as protective but again more context is needed-is this because of toxicity/first line failure etc. Why is this the case? - In the conclusion, late presentation is included as a concern, however this does not come through clearly in the results, does this mean enrolment in later years (this is fairly clear in results) or late initiation in children? Please clarify Introduction  - Well written Methods  - Clearly written - Line 22 page 7, why are only children > 5 included in the advanced disease definition? Even though this definition would not have influenced treatment start at the time of the study, it would be important to know how many children under 5, who should be most reliably diagnosed through the PMTCT program, present late with advanced disease. Results  - The tables are a bit difficult to interpret, specifically the percentage columns for anemia, WAZ etc, do not add to 100% although categorical variables are included (anemia yes/no) - Lines 41-46 needs elaboration-why did patients change their ART-failure/toxicity? Why was this protective? - Similar comments for percentages for the remaining tables - Table 2 should be clearly labelled with a heading at the top - How was adherence calculated-this is not discussed in methods Discussion  - Although well written, the discussion is way too long and could be considerably shortened. - The discussion should more clearly focus on major findings upfront, and tailor discussion accordingly. - PMTCT sections could be summarised, discussing the key points around why less children are infected with HIV in recent years - Cited attrition data can be substantially shorted, too much detail is included. A summary would suffice - Please discuss reasons for treatment switch and how this finding
--

	being protective compares to other data - As a limitation (rather than strength) the sample size is actually very small Conclusion - Re-discusses many of the results and includes IQR. Should only highlight key findings and how this impacts paediatric HIV care. - Should not be more than 1 paragraph
--	--

VERSION 1 – AUTHOR RESPONSE

To BMJ Paediatrics Open Editorial Office We would like to thank Editor-in-chief, and Associate Editor of BMJ Paediatrics Open and our reviewers for their invaluable inputs and constructive comments that are helpful to massively improve the quality of the manuscript. After careful consideration of the points raised, the point-by-point response are as follows: A. Response to “Editor in Chief Comments to Author”: B. Response to “Associate Editor Comments to Author”:

Points raised or amendments requested by Associate Editor Authors’ response 1. The results and discussion sections should be made more succinct, and some of the statistics used need greater clarification. Attempts have been made to make the discussion more succinct. And effort has been made to clarify some of the statistics used. Otherwise see relevant comments to reviewer one and two for more details. Overall, 943 revisions have been made. These includes: 266 insertions, 226 deletions, 10 moves, and 441 formatting. Points raised or amendments requested by Editor in Chief Authors’ Response 1. Title amends to "Attrition and associated factors among HIV-infected children at a tertiary hospital in Eritrea: A retrospective cohort analysis" Title has been overhauled 2. Introduction needs shortening The section has been shorted from 552 to < 476 words. 3. Discussion deletes the 1st sentence. Journal policy is for authors to avoid describing their study as the first This comment is addressed in the revised manuscript. 4. Strengths and limitations section. Delete the 1st sentence. Journal policy is for authors to avoid describing their study as the first. This comment is addressed in the revised manuscript. 5. Conclusions. Delete the 1st sentence. Journal policy is for authors to avoid describing their study as the first. This comment is addressed in the revised manuscript. 6. Discussion. Dramatically shorten. The discussion has been overhauled. Entire sections have been deleted or rewritten. In the process, the word count has been reduced from 2,249 to ~1,424. 7. Do not shorten the methods or results Directive noted Page | 2 C. Response to Reviewer-1 (Dr. Rachel Mary Hilliam, The Open University): Points raised or amendments requested by reviewer-1 Authors’ response 1. The first is what exactly the p-values relate to in Table 1. Are these chi-squares for total numbers across different years (i.e., male/female three-year groups), in other words treating this as a 3x2 contingency table? This is not entirely clear if this is the case in your results section which discusses the baseline characteristics as most of this section repeats what is given in the table rather than adding an explanation to the results. In table 1, p-value represent the likely hood of the test results (proportions/means/median) assuming null hypothesis (Ho) is true, i.e., there is no difference of proportions/means/median of the stated variables across the different bands of calendar years. Furthermore, it has now been overhauled to make sure all the pvalues are explained by superscripts and elaborative foot notes. 2. Similarly, in Table 2 you need to be clear what these p-values relate to. In this case are you comparing number of Attrition with number of retentions in each of the categories I think the different years of attrition and retention also need some explanation, particularly in terms of what this means for the data. The Abbreviations for statistical tests for difference between retention and attrition against specific cohort characteristics: χ^2 Chi-square test;

a; Mann-Whitney u test, b; independent samples t-test. Yes, the comparisons are for number of attritions with number of retentions in each of the categories. 3. In Table 3 I'm assuming the p-values here from the proportional hazards? This again isn't clear. The following statement has been added at the bottom of the table: p-values are for mean survival duration in years. 4. In Table 4 and throughout the paper it is usual to write confidence intervals with the lower and upper limit in round brackets, not with separating the two limits. So (LL, UL) would be standard statistical notation. We evaluated your opinion and agree that this is a possible notation. However, we feel that most attrition and retention studies we reviewed employ this notation. See Collins IJ, Jourdain G, Hansudewechakul R, et al. Long-Term Survival of HIV-Infected Children Receiving Antiretroviral Therapy in Thailand: A 5-Year Observational Cohort Study. *Clinical Infectious Diseases* 2010; 51(12):1449–1457. We also noted that the journal entertains a range of styles – see Bimer KB, Sebsibe GT, Page | 3 Desta KW, et al. Incidence and predictors of attrition among children attending antiretroviral follow-up in public hospitals, Southern Ethiopia, 2020: a retrospective study. *BMJ Paediatrics Open* 2021;5:e001135. doi:10.1136/bmjpo-2021-001135 Therefore, we have decided to retain it. However, we are willing to comply with this directive. 5. I would also suggest that you need much more context around the results in Table 4 for the readers of this journal. The results reported in this table are discussed in the last three paragraphs of the discussion. In the revised version of the paper, we have added more details and context. 6. I would suggest a slight restructuring and putting some of the context which is in the discussion into the appropriate subsections of the results. This would help readers understand exactly what is being analyzed and put the results in context for each of the Tables. This advice has been headed. The following statements are included in the result section. In text description of table 1 - The following description has been added under table 2 - Abbreviations for statistical tests for difference between retention and attrition against specific cohort characteristics: χ Chi-square test; a; Mann-Whitney u test, b; independent samples t-test. Description of table 3 - Kaplan-Meier survival curves with log rank tests. In text description of table 4 - The hazard ratios (unadjusted and adjusted) for attrition in 701 patients and associated p-values for difference across specific cohort characteristics are given in Table 4. Page | 4 D. Response to Reviewer-2 (Prof. Lee Fairlie, Wits RHI): Points raised or amendments requested by reviewer-2 Authors' response ABSTRACT 1. Suggest using children living with HIV (CLHIV) or HIV-positive children throughout and remaining consistent The revised manuscript has now children living with HIV (CLHIV) throughout the context in place of HIV-positive or HIV-infected children. 2. Line 24 "Among the 71 0....."seems to stop short and should be revised This statement has been amended. 3. The manuscript repeatedly refers to "residence outside Zoba Maekel" which seems to predict worse outcomes; however, this is not very clear for people unfamiliar with this city/region so please elaborate on why this may be important-is it more rural/less expertise/fewer children and therefore less confidence etc. The following statement has been added in the operational definition section: Zoba Maekel is one of the administrative zones in Eritrea. Therefore, residence outside Zoba Maekel refers to children who reside outside this administrative zone. In addition....the following sentence has been added to the discussion section.... For context, ridicule from friends/peers in particular; may force guardians/parents to enroll patients in distant facilities – residence outside Zoba Maekel. 4. Please spell out in full BAZ/WAZ/HAZ and other abbreviations the first time they're used. This is as well addressed in the revised manuscript e.g. 5. In the conclusion, late presentation is included as a concern, however this does not come through clearly in the results, does this mean enrolment in later years (this is fairly clear in results) or late initiation in children? Please clarify By 'late presentation', we mean patients which are getting enrolled with advanced WHO HIV stage (III/IV)

and/or T-cell CD4+ count less than 200 cells/ μ L. Importantly, the following line has been added in the section on operational definition - Late presentation was defined as patients enrolled with advanced WHO HIV stage (III/IV) and/or T-cell CD4+ count less than 200 cells/ μ L. ...see the following reference for additional clarification... Antinori A, Coenen T, Costagiola D, Dedes N, Ellefson M, Gatell J et al. Late Presentation of HIV Infection: A Consensus Definition. *HIV Med.* 2011;12:61–4.

METHOD 1. Line 22 page 7, why are only children > 5 included in the advanced disease definition? Even though this definition would not have influenced treatment start at the time of the study, it would be important to know how many children under 5, who should be most reliably diagnosed through the PMTCT program, present late with advanced disease. According to the new WHO Combined Antiretroviral therapy guideline, all children < 5 years who are HIV infected are assumed to have advanced HIV disease as they would benefit from approach used to treat individuals with advanced HIV disease. Please, look into the below cited reference for further clarification: WHO: Guidelines for managing advanced HIV disease and rapid initiation of antiretroviral therapy, July 2017. RESULTS 1. The tables are a bit difficult to interpret, specifically the percentage columns for anemia, WAZ etc, do not add to 100% although categorical variables are included (anemia yes/no) Appropriate changes have been made; also see comments addressed to other reviewers. 2. Lines 41-46 needs elaboration-why did patients change their ART failure/toxicity? Why was this protective? An explanation has been provided in the final paragraph of the discussion. 'Note that retention in care may correlate with a higher likelihood of cART changes/switch to other backbones. In contrast, late presentation is associated with higher likelihood of mortality/attrition within the first 6 months of enrollment³². This can in turn undercut the prospects of cART switches particularly in settings with a limited range of cART option. If valid, then the latter explanation highlights a major systematic weakness in cART clinics across Eritrea – the limited diversity of cART options for children CLHIV and ALHIV. ' 3. Similar comments for percentages for the remaining tables Appropriate changes have been made in the revised document. Page | 6 4. Table 2 should be clearly labelled with a heading at the top Title of table 2 has been placed below during formatting the manuscript, but now it's placed in the top of the table. 5. How was adherence calculated-this is not discussed in methods Adherence was assessed at each follow-up visit as good, fair, and poor if a child missed 10% doses respectively of the expected monthly doses. This statement is included in the method section under the subsection Operational definition. Discussion 1. Although well written, the discussion is way too long and could be considerably shortened. The discussion has been overhauled. Entire sections have been deleted or re-written. In the process, the word count has been reduced from 2,249 to ~1,424. 2. The discussion should more clearly focus on major findings upfront, and tailor discussion accordingly Appropriate changes have been made... entire sections have been removed and several sections rewritten to comply with this directive. 3. PMTCT sections could be summarized, discussing the key points around why less children are infected with HIV in recent years This section has been rewritten...See Actually ...the low enrollment numbers in 2016 to 2020 may simply be a reflection of the fact that PMTCT programs in Asmara are closer to elimination of mother-to-child transmission (MTCT) ... Whatever the case, we can cautiously conclude that the data highlights the success of HIV programs in Asmara, Eritrea. 4. Cited attrition data can be substantially shortened, too much detail is included. A summary would suffice The directive has been implemented. The paragraph has been shortened considerably e.g.This incidence estimate is lower when compared to studies conducted in Ethiopia (8.36[95% CI: 7.12–9.80]/100 PY) 19; Zimbabwe (11.8, [95% CI:11.0– 12.7])/100 PY 20; Nigeria (10.8/100 PY) 21 and Kenya (23.1/100 PY) 22 . In contrast, the incidence of attrition was higher than what was reported elsewhere^{23 24 25} 6. As a limitation (rather than strength) the

sample size is actually very small Appropriate revisions have been undertaken. Page | 7 Conclusion
1. Re-discusses many of the results and includes IQR. Should only highlight key findings and how this impacts paediatric HIV care. (Should not be more than 1 paragraph) The conclusion section has been restructured and shortened considerable. The current word count is 199 words.

VERSION 2 – AUTHOR RESPONSE

What the study adds delete the first statement The first statement has been deleted
Results need expanding. 172 were lost to attrition. You MUST state the number who died in the text (in the Results, not Discussion) and the numbers who were lost to follow up. The suggested statement has been added in the results section.

Please state statistical test before P values in Results: see Clinical and disease-related characteristics of CLHIV at baseline section. All statistical tests are named before p values in the results now.

Discussion deletes the first 3 paragraphs. Your paper is about attrition. All the recommended amendments have been made.

Discussion deletes the Conclusion section. It does not help the paper. All the recommended amendments have been made.

The English need improving before resubmission. BMJ will do this. Please contact our office The editage author service has made the edits. You can find the changes in the new submissions tracked changes.